# ArchPower: Dataset for Architecture-Level Power Modeling of Modern CPU Design

**Qijun Zhang, Yao Lu, Mengming Li, Shang Liu, Zhiyao Xie**[*]
Hong Kong University of Science and Technology
{qzhangcs, yludf, mengming.li, sliudx}@connect.ust.hk, eezhiyao@ust.hk

## Abstract

Power is the primary design objective of large-scale integrated circuits (ICs), especially for complex modern processors (i.e., CPUs). Accurate CPU power evaluation requires designers to go through the whole time-consuming IC implementation process, easily taking months. At the early design stage (e.g., architecture-level), classical power models are notoriously inaccurate. Recently, ML-based architecture-level power models have been proposed to boost accuracy, but the data availability is a severe challenge. Currently, there is no open-source dataset for this important ML application. A typical dataset generation process involves correct CPU design implementation and repetitive execution of power simulation flows, requiring significant design expertise, engineering effort, and execution time. Even private in-house datasets often fail to reflect realistic CPU design scenarios. In this work, we propose ArchPower, the first open-source dataset for architecture-level processor power modeling. We go through complex and realistic design flows to collect the CPU architectural information as features and the ground-truth simulated power as labels. Our dataset includes 200 CPU data samples, collected from 25 different CPU configurations when executing 8 different workloads. There are more than 100 architectural features in each data sample, including both hardware and event parameters. The label of each sample provides fine-grained power information, including the total design power and the power for each of the 11 components. Each power value is further decomposed into four fine-grained power groups: combinational logic power, sequential logic power, memory power, and clock power. ArchPower is available at https://github.com/hkust-zhiyao/ArchPower.

## 1 Introduction

The rapid advancements of AI rely on the support of very large-scale integrated (VLSI) circuits. *Power* is the primary design objective of integrated circuits (ICs), especially for complex modern processors (i.e., CPUs), which play a central role in various computing systems. Accurate yet efficient power estimation techniques are the premise of and key challenge of power optimization. However, as Fig. 1(a) shows, accurate CPU power evaluation requires designers to go through the whole time-consuming IC implementation process, easily taking months. As the complexity of CPU designs keeps increasing, the standard power estimation flow becomes increasingly costly.

To facilitate power estimation at early stages, designers will evaluate power consumption at the architecture level, before designing the RTL (e.g., in Verilog or VHDL) and going through the downstream implementation flow (i.e., circuit synthesis and layout). Fig. 1(b) illustrates the workflow of the architecture-level power modeling, using classical tools such as McPAT [14] and Wattch [10]. Such a fast power estimation approach takes only tens of seconds, which is about $100\times$ faster than the

---

[*]Corresponding Author

39th Conference on Neural Information Processing Systems (NeurIPS 2025) Track on Datasets and Benchmarks.

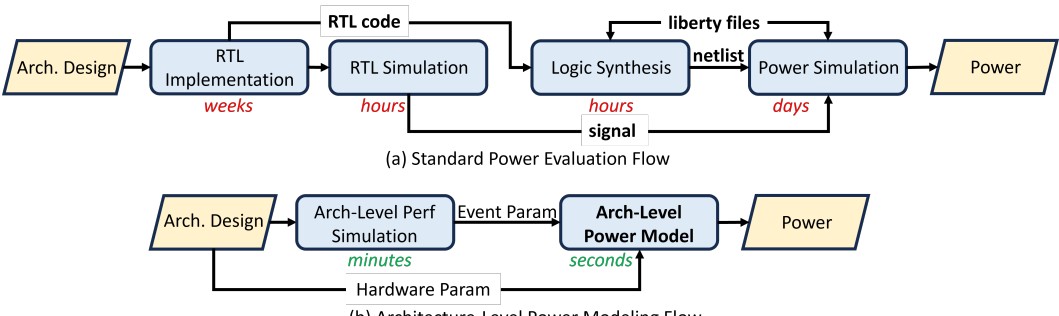

(a) Standard Power Evaluation Flow

(b) Architecture-Level Power Modeling Flow

Figure 1: Comparison between (a) standard power evaluation flow and (b) architecture-level power evaluation flow. The architecture-level power modeling flow is significantly efficient compared with the standard power evaluation flow. ArchPower provides labeled data for ML-based architecture-level power modeling.

| Work | Commercial Tech Lib | Clock Gating | SRAM Implementation | Diverse Architectures |
|------|:---:|:---:|:---:|:---:|
| McPAT-Calib [20] | | | | |
| ASPDAC'23 [22] | | | | |
| PANDA [24] | ✓ | | ✓ | |
| FirePower [23] | ✓ | | ✓ | ✓ |
| AutoPower [25] | ✓ | ✓ | ✓ | |
| **ArchPower (This Work)** | ✓ | ✓ | ✓ | ✓ |

Table 1: Comparison between datasets in existing works and our proposed ArchPower dataset.

standard VLSI power estimation flow. However, these classical analytical architecture-level power models are notoriously inaccuracy, as indicated in multiple existing studies [17, 20, 13, 16, 12].

In recent years, machine learning (ML)-based architecture-level power model [20, 24] has been explored for better power evaluation accuracy. The ML-based architecture-level power model takes both *hardware parameters* and *event parameters* as features to predict the CPU power consumption as its output. Hardware parameters are parameters to determine CPU configurations, such as *FetchWidth* and *DecodeWidth*. Event parameters are event statistics when a CPU executes a workload, collected from existing architecture-level performance simulators, such as the number of branch mispredictions and DCache misses. Based on a few training data collected on the target CPU architecture, ML-based power models can mitigate the modeling error or bias incurred from analytical models that are built for outdated processors.

However, despite emerging works in ML-based architecture-level power models [20, 22, 24, 23], they all built their solutions on private datasets. There is no open-source dataset for such an important application, preventing the AI community from making its contribution. We find that some of them open-source their model implementation [20, 24, 23], however, none of them open-source their dataset for training and testing. Other related works, such as architecture-level design space exploration [7, 8, 19, 21, 6], also do not share their data. It is because the dataset generation for the ML-based architecture-level power modeling is challenging, requiring significant IC knowledge and engineering effort for the correct CPU design implementation and power simulation flow.

Besides the unavailability, these in-house datasets also have other limitations, as shown in Table B. 1) Some datasets [20, 22] do not include SRAM in their implementation. It is because of difficulties in implementing SRAM in RTL and the lack of SRAM support in some technologies. However, the SRAM is essential to build many important components of the CPU, such as the cache and the branch predictor, and consumes over 50% power of the whole CPU. Therefore, the absence of SRAM leads to an unreliable evaluation. 2) Some other datasets [24, 23] do not adopt the clock-gating technique for logic synthesis, making the clock power far from the real processors. 3) Most of these datasets [20, 22, 24] are only collected based on a single CPU architecture, unable to validate whether the evaluated models can also work for other architectures.

To address the problems above, in this work, we propose ArchPower, the first open-source dataset for ML-based architecture-level power modeling of modern processor design. Our dataset includes 200 samples collected from 25 CPU configurations and 8 workloads. The architectural feature of each sample is a vector with 101 elements, including hardware parameters and event parameters. They can also be extracted as per-component features with our provided indexes. The power label of

each sample includes the whole CPU ground-truth power and 11 per-component ground-truth powers. Each ground-truth power has not only the total circuit power but also the fine-grained power values of four power groups, including combinational logic power, sequential logic power, memory power, and clock power.

To build the dataset, we invest substantial engineering effort. We set up the frameworks [4, 18] for the RTL code generation process of two widely adopted CPU architectures, including BOOM [26] and XiangShan CPUs [18]. We also integrate realistic SRAM macros for each memory block in the RTL designs. Based on the RTL implementation, we go through the complex VLSI flow and standard power evaluation flow with commercial EDA tools [3, 15, 2], with clock-gating considered. The VLSI flow consumes a long runtime and significant computing power.

Our contributions are summarized below.

- We release ArchPower, the *first* open-source dataset for the ML-based architecture-level power model. ArchPower includes 200 data samples collected from 25 CPU configurations and 8 workloads. ArchPower reflects realistic design power since it considers the clock-gating and integrates realistic SRAM macros for power label collection.

- We also provide a training-testing framework for the ML-based architecture-level power model. It includes different setups of training and testing data that reflect the realistic CPU development scenarios, which can evaluate the generalization of ML-based power models.

- We evaluate six different power models, including two analytical models and four ML-based models, based on ArchPower. The evaluation provides both the total power and per-component power modeling accuracy.

## 2 Preliminary

In this section, we first describe the principle of the standard power evaluation flow in Sec. 2.1, briefly introduce both classical analytical architecture-level power model and ML-based architecture-level power model in Sec. 2.2, and then introduce the modern CPU architecture in Sec. 2.3.

### 2.1 Principle of VLSI Power Evaluation Flow

The dynamic power dominates the power consumption, and the leakage power is small compared with the dynamic power. Therefore, we focus on the dynamic power evaluation. The dynamic power of the processor is the sum of the dynamic power of all cells. Denoting the dynamic power as $P$, the power calculation is shown in Eq.(1),

$$P = \sum_{c \in netlist} \alpha_c C_c V^2 f \tag{1}$$

where $c$ is the cell in the netlist, $\alpha_c$ is the switch activity of cell $c$, $C_c$ is the capacitance of cell $c$ and the load capacitance of its wire, $V$ is the voltage of the chip, and $f$ is the frequency of the chip.

The standard power evaluation flow calculates the power by collecting all related values from inputs and conducting the calculation. The cell $c$ is from the netlist generated by logic synthesis, switch activity $\alpha_c$ is extracted from the signal information generated by RTL simulation, capacitance $C_c$ is from the liberty file, and the voltage $V$ and frequency $f$ are set at the chip level. Such a standard power calculation is complex and slow because both generating input files and calculating the final power are time-consuming.

### 2.2 Architecture-Level Power Model

To avoid the time-consuming standard power evaluation, the architecture-level processor power model can provide fast power estimation at the early design stage. The architecture-level power model takes the hardware parameters $H$, which determine the CPU configurations, and event parameters $E$, which are the event statistics generated by the performance simulators like gem5 [9], as input and outputs the power consumption $P$.

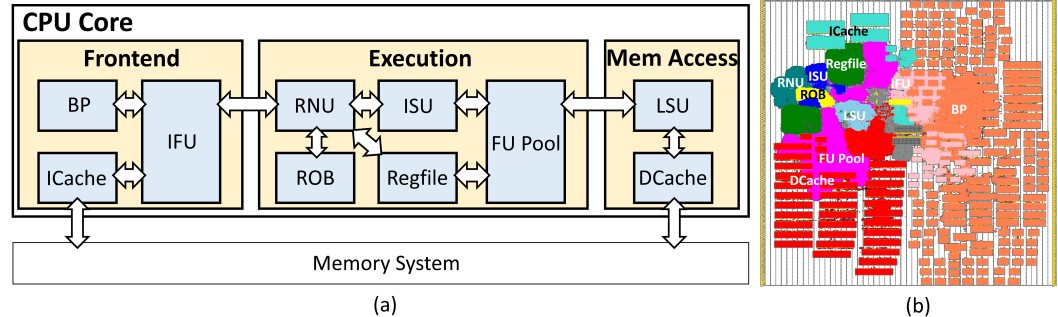

Figure 2: (a) The architecture of the modern high-performance CPU core. Blue blocks are major components. The yellow block represents the Other Logic. (b) A layout example of one BOOM CPU.

**Analytical Architecture-Level Power Model:** The analytical architecture-level power model estimates the processor power consumption following two steps, denoted as Eq.(2).

$$P = F_{event}(F_{op}(H), E) \qquad (2)$$

In the first step, based on the hardware parameter $H$, the model instantiates each component of the processor based on empirical models to collect the per-operation energy. This step is denoted as $F_{op}$. In the second step, the model transforms the event parameters $E$ into the count of basic operations and calculates the power consumption. This step is denoted as $F_{event}$. However, because of the discrepancy between the real processor and the modeled one, the analytical $F_{op}$ and $F_{event}$ are usually inaccurate, leading to the low accuracy of analytical power models.

**ML-based Architecture-Level Power Model:** In recent years, to address the low accuracy of analytical power models, ML-based architecture-level power models [20, 22, 24, 23] have been proposed. The ML-based power model adopts a machine learning model $F_{ml}$ to directly learn the correlation between the input feature and the final power consumption, denoted as Eq.(3).

$$P = F_{ml}(H, E) \qquad (3)$$

Among the ML-based architecture-level power models, some existing ML-based power models [20, 22] adopt a purely black-box machine learning algorithm, and some others [24, 23] utilize a hybrid gray-box model, with some analytical information provided.

### 2.3   Modern CPU Architecture

Fig. 2 shows the basic architecture of modern high-performance CPUs that our dataset targets. Modern CPUs usually adopt out-of-order execution to improve instruction-level parallelism, which can significantly boost CPU performance. The CPU has three major blocks, including Frontend, Execution, and Mem Access, with each block consisting of multiple components. 1) The Frontend includes 3 components: branch predictor (BP), instruction cache (ICache), and instruction fetch unit (IFU). 2) The Execution consists of 5 components: renaming unit (RNU), reorder buffer (ROB), issue unit (ISU), register file (Regfile), and function unit pool (FU Pool). 3) The Mem Access has 2 components: load-store unit (LSU) and data cache (DCache). 4) The logic not covered by the major components above is referred to as Other Logic. ArchPower provides per-component power labels for each of the 11 components.

## 3   Related Work

Despite many existing works exploring architecture-level power modeling, there is *no* existing open-source dataset for the architecture-level power model. Some existing works [20, 22, 24, 23] release their model implementation code, and some of them [20, 22] also provide example data for demonstration with only one or two samples. However, none of them release their full dataset for training and testing.

Besides unavailability, these in-house datasets also have other problems. Some works [20, 22] exclude the SRAM in the processors, while the SRAM is the basic building block of many important components and dominates the power consumption of modern CPUs. Some other works [24, 23]

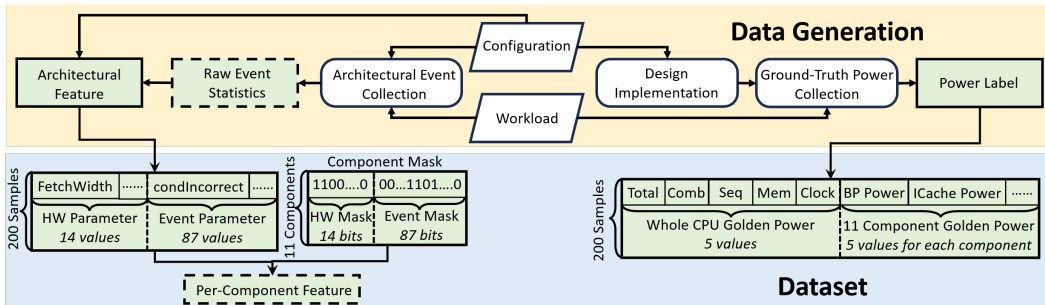

Figure 3: Dataset and data generation process of ArchPower. Our dataset mainly includes the architectural power modeling features and power labels. The features can further be masked with component masks to generate per-component features. ArchPower generates architectural power modeling features through the architectural event collection and generates power labels through design implementation and ground-truth power collection.

do not adopt the clock-gating technique when performing logic synthesis, which is an essential optimization for processor designs. Therefore, ignoring the clock-gating technique makes their CPU implementation and power labels far from the real processors. Therefore, an open-source high-quality dataset is in great need for the development of ML-based architecture-level power models.

## 4 Dataset Description

### 4.1 Dataset Overview

Our ArchPower dataset consists of $25 \times 8 = 200$ data samples collected from 25 CPU configurations when executing 8 different workloads. Each data sample describes a CPU configuration when executing a workload, providing both architecture-level *features* and power *labels*. Fig. 3 provides an overview of our ArchPower dataset. The architectural feature of each sample is a vector with 101 elements, including 14 hardware parameters $H$ and 87 event parameters $E$. The corresponding ground-truth power label of each sample is collected by going through the design implementation and simulation flow.

In addition to the feature and power label of the whole design, we provide component masks to indicate per-component features and include per-component power labels for 11 components. For both the whole design and per-component, we also provide fine-grained power labels for four power groups: combinational logic power, sequential logic power, memory power, and clock power. The power label of each sample has 60 values in total. The raw event statistics files of each sample are also included for potential use in future research.

### 4.2 Detailed Dataset Description

Given the hardware parameter $H$ of the target CPU configuration and event parameter $E$ when the CPU runs the target workload, the ML-based architecture-level power model predicts the power consumption $P$, as shown in Eq.(3). Therefore, in ArchPower, a sample represents a CPU configuration running a workload. ArchPower provides features and labels for both the whole CPU and each component. In this subsection, we describe the feature and label in detail.

#### 4.2.1 Architectural Power Modeling Feature

ArchPower has 200 data samples, and each sample has 101 features, including 14 hardware parameters and 87 event parameters. Therefore, we provide the architectural power modeling feature as a $200 \times 101$ matrix in our dataset. 1) The first 14 columns are the 14 hardware parameters that determine the CPU configuration, including *FetchWidth*, *DecodeWidth*, *FetchBufferEntry*, *RobEntry*, *IntPhyRegister*, *FpPhyRegister*, *LDQ/STQEntry*, *BranchCount*, *Mem/FpIssueWidth*, *IntIssueWidth*, *DCache/ICacheWay*, *DTLBEntry*, *MSHREntry*, and *ICacheFetchBytes*. 2) The last 87 columns are the 87 event parameters that are event statistics generated by the architecture-level performance simulator when a CPU configuration executes a workload, such as *condIncorrect*, *icache.overallMisses*, and *dcache.ReadReq.access*. Table 2 lists hardware parameters and event parameters of each component.

| Component | Hardware parameters of each component | Event parameters of each component |
|---|---|---|
| BP | FetchWidth, BranchCount | BTBLookups, condPredicted, condIncorrect, commit.branches |
| IFU | FetchWidth, DecodeWidth FetchBufferEntry, ICacheFetchBytes | fetch.{insts, branches, cycles}, numRefs, numStoreInsts, numInsts, decode.{runCycles, blockedCycles, decodedInsts}, numBranches, intInstQueueReads, intInstQueueWrites, intInstQueueWakeupAccesses, fpInstQueueReads, fpInstQueueWrites, fpInstQueueWakeupAccesses |
| ICache | ICacheWay, ICacheFetchBytes | overallAccesses, overallMisses, ReadReq.mshrHits, ReadReq.mshrMisses, tagAccesses |
| RNU | DecodeWidth | intLookups, renamedOperands, fpLookups, renamedInsts, runCycles, blockCycles, committedMaps |
| ROB | DecodeWidth, RobEntry | reads, writes |
| ISU | DecodeWidth, Mem/FpIssueWidth, IntIssueWidth | IssuedMemRead, IssuedMemWrite, IssuedFloatMemRead, IssuedFloatMemWrite, IssuedIntAlu, IssuedIntMult, IssuedIntDiv, IssuedFloatMult, IssuedFloatDiv |
| Regfile | DecodeWidth, IntPhyRegister, FpPhyRegister | intRegfileReads, fpRegfileReads, intRegfileWrites, fpRegfileWrites, functionCalls |
| FU Pool | Mem/FpIssueWidth, IntIssueWidth | intAluAccesses, fpAluAccesses |
| LSU | LDQ/STQEntry, MemIssueWidth | MemRead, InstPrefetch, MemWrite |
| DCache | DCacheWay, DCacheTLBEntry, DCacheMSHR, MemIssueWidth | ReadReq.accesses, WriteReq.accesses, ReadReq.misses, tagAccesses, WriteReq.misses, overallMisses, MshrHits, MshrMisses |
| CPU Level | - | totalIpc, totalCpi, numCycles, idleCycles, numLoadInsts, numSquashedInsts, committedInsts, commit.{numDist::mean, memRefs}, mmu.dtb.{accesses, misses}, iew.writebackCount, numIssuedDist::mean, statIssuedInstType_0::total, fuBusy, mmu.itb.{accesses, misses}, conflictingLoads, conflictingStores, insertedLoads, insertedStores, mem_ctrls.{readReqs, writeReqs, bytesReadSys}, icache.tags.totalRefs, dcache.{overallAccesses::total, tags.totalRefs} |

Table 2: Hardware parameters and event parameters of each component. The 14 hardware parameters and 87 event parameters in the architectural power modeling feature are the union of all components and the CPU-level parameters. Other Logic adopts all features and is not listed.

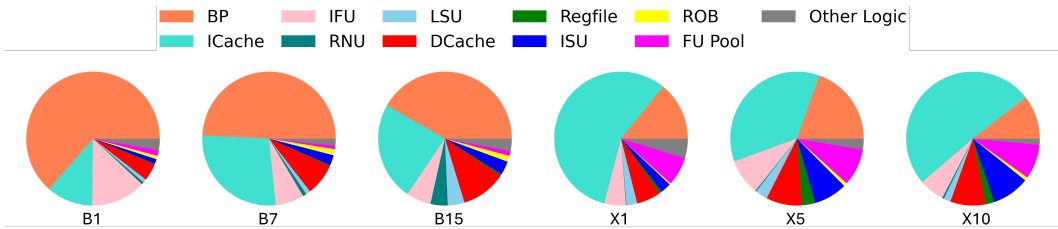

Figure 4: The power distributions across 11 components of 6 different CPU configurations (B1, B7, B15, X1, X5, X10) with different scales.

The 14 hardware parameters and 87 event parameters in the architectural power modeling feature are the *union* of all components and the CPU-level parameters.

Our dataset provides a component mask to select the features for each component, as shown in Fig. 3. The Other Logic adopts all features and is not listed. For each component, the component mask has a 14-bit hardware mask to select from 14 hardware parameters and an 87-bit event mask to select from 87 event parameters. With the component mask, the per-component features can be extracted from our provided architectural power modeling feature.

### 4.2.2 Power Label

For the 200 data samples, each sample has $(1 + 11) \times (1 + 4) = 60$ fine-grained power labels, for both the whole CPU and 11 components. We further decouple the power into four power groups, including combinational logic power, sequential logic power, memory logic power, and clock power. Therefore, we provide our power label as a $200 \times 60$ matrix in our dataset. For each sample, the label includes the whole CPU ground-truth power collected from the standard VLSI power evaluation flow. Besides the whole CPU ground-truth power, we also provide the labels at the component level, which include ground-truth power for the 11 components, including BP, IFU, ICache, RNU, ROB, ISU, Regfile, FU Pool, LSU, DCache, and Other Logic, respectively. Fig. 4 shows the power distribution across different components for configurations with different scales, where Bi is the $i^{\text{th}}$ configuration of BOOM architecture and Xi is the $i^{\text{th}}$ configuration of XiangShan architecture.

| Hardware Parameter | B1 | B2 | B4 | B6 | B7 | B9 | B11 | B13 | B15 | X1 | X3 | X5 | X7 | X8 | X10 |
|---|---|---|---|---|---|---|---|---|---|---|---|---|---|---|---|
| FetchWidth | 4 | 4 | 4 | 8 | 8 | 8 | 8 | 8 | 8 | 4 | 4 | 4 | 8 | 8 | 8 |
| DecodeWidth | 1 | 1 | 2 | 2 | 3 | 3 | 4 | 5 | 5 | 2 | 2 | 3 | 4 | 4 | 5 |
| FetchBufferEntry | 5 | 8 | 8 | 24 | 18 | 30 | 32 | 30 | 40 | 8 | 24 | 24 | 24 | 32 | 24 |
| RobEntry | 16 | 32 | 64 | 80 | 81 | 114 | 128 | 125 | 140 | 16 | 48 | 64 | 81 | 96 | 112 |
| IntPhyRegister | 36 | 53 | 64 | 88 | 88 | 112 | 128 | 108 | 140 | 36 | 68 | 80 | 88 | 110 | 108 |
| FpPhyRegister | 36 | 48 | 56 | 72 | 88 | 112 | 128 | 108 | 140 | 36 | 68 | 80 | 88 | 110 | 108 |
| LDQ/STQEntry | 4 | 8 | 12 | 20 | 16 | 32 | 32 | 24 | 36 | 16 | 24 | 24 | 24 | 32 | 32 |
| BranchCount | 6 | 8 | 10 | 14 | 14 | 16 | 20 | 18 | 20 | 7 | 7 | 7 | 7 | 7 | 7 |
| Mem/FpIssueWidth | 1 | 1 | 1 | 1 | 1 | 2 | 2 | 2 | 2 | 2 | 2 | 2 | 2 | 2 | 2 |
| IntIssueWidth | 1 | 1 | 1 | 2 | 2 | 3 | 4 | 5 | 5 | 2 | 2 | 4 | 4 | 6 | 6 |
| DCache/ICacheWay | 2 | 4 | 4 | 8 | 8 | 8 | 8 | 8 | 8 | 4 | 8 | 4 | 8 | 8 | 8 |
| DTLBEntry | 8 | 8 | 8 | 16 | 16 | 32 | 32 | 32 | 32 | 8 | 16 | 8 | 16 | 16 | 32 |
| MSHREntry | 2 | 2 | 2 | 4 | 4 | 4 | 4 | 8 | 8 | 2 | 4 | 2 | 4 | 4 | 4 |
| ICacheFetchBytes | 2 | 2 | 2 | 4 | 4 | 4 | 4 | 4 | 4 | 2 | 2 | 2 | 2 | 2 | 2 |

Table 3: Some representative CPU configurations across different scales from our dataset. B1-B15 denote 15 configurations of BOOM, and X1-X10 denote 10 configurations of XiangShan.

## 4.3 Training-Testing Data Setup

In addition to the dataset, we also provide a ready-for-use training-testing framework for the evaluation of ML-based architecture-level power models. In our framework, we set up training scenarios based on the three unique characteristics of processor developments. 1) Training and testing samples are divided based on configurations, where all data collected from training configurations are training data, and data from testing configurations are testing data. 2) Because of the significant manpower and time overhead, the training configurations are usually limited in real scenarios. Therefore, we set up few-shot scenarios with only three training configurations. 3) In the industry, architects usually also work on configurations that have different scales from available training configurations. Therefore, we set up three training scenarios with different training data distributions. Therefore, we set up three training-testing scenarios named *Balance*, *Small*, and *Large*: 1) *Balance*. We evenly select the configurations as available training configurations based on the scale: B1, B8, and B15 for BOOM, X1, X6, and X10 for XiangShan. 2) *Small*. We select the smallest configurations as available training configurations: B1, B2, and B3 for BOOM, X1, X2, and X3 for XiangShan. 3) *Large*. We select the largest configurations as available training configurations: B13, B14, and B15 for BOOM, X8, X9, and X10 for XiangShan.

Besides directly utilizing the training-testing data setup in our provided evaluation framework, users can also evaluate their own training-testing data setup based on the provided dataset of ArchPower.

# 5 Dataset Generation Process

## 5.1 Adopted CPU Configurations and Workloads

We adopt two highly configurable CPU architectures, BOOM [26] and XiangShan [18], to generate multiple CPUs with different configurations. The Berkeley Out-of-Order Machine (BOOM) [26] implemented in Chisel [5] is a synthesizable and parameterizable open-source out-of-order core. It can be configured to cores with different scales, given a variety of hardware parameters $H$. Xiang-Shan [18] is a high-performance open-source CPU project. Similar to the BOOM, the XiangShan is also implemented with Chisel and is highly configurable. In our dataset, we adopt 15 configurations of the BOOM CPU named B1-B15. We adopt 10 configurations of the XiangShan CPU in our dataset named X1-X10. The CPU configurations that we adopted are carefully selected to be similar to real-world commercial CPUs. Different hardware parameters within each configuration also configure a CPU where components are balanced. Some representative CPU configurations are listed in Table 3 because of the page limitation. All CPU configurations are provided in the appendix.

For the workloads executed on the CPU, we utilize workloads from the riscv-tests [1]. Riscv-tests is the official test benchmark for the RISC-V processors. We collect 8 widely adopted real-world workloads from riscv-tests, including dhrystone, median, multiply, qsort, rsort, towers, spmv, and vvadd. These workloads are across different lengths, from thousands of cycles to several hundred thousand cycles.

### 5.2 Data Collection Flow

**Architectural Event Collection:** For a CPU configuration when executing a workload, we adopt the gem5 [9] as our performance simulator to generate the event statistics. We configure the O3CPU in gem5 with the hardware parameters of the simulated configuration and execute the workload. All of our generated raw event statistic files are available in our dataset, and our script for automatic configuration and simulation is also open-sourced in ArchPower.

**Design Implementation:** To implement a CPU design with a configuration, we perform RTL code generation with Chipyard [4] v1.8.1 and OpenXiangShan [18] for BOOM and XiangShan, respectively. To get the netlist, we perform logic synthesis with Synopsis Design Compiler® [2], during which is clock-gating technique is turned on. The technology library utilized in our implementation is 40nm standard cell library. We also implement the SRAM in the processor using the Memory Compiler of the 40nm technology library.

**Ground-Truth Power Collection:** For a configuration executing a workload, we perform the standard power evaluation flow to collect the ground-truth power. We perform RTL simulation with Synopsys VCS® [3]. We perform post-synthesis power simulation with PrimePower [15] and collect the power data as the label.

**Advanced Technology Library:** Besides the primary 40nm technology library on which our experiment is performed in this paper, we also provide an additional dataset collected with a 28nm technology library. This additional dataset has the same organization as our primary 40nm dataset, and can be adopted for benchmarking with the same benchmark framework. In the future, if we get access to any high-quality FinFET technology library in the future, we will update our dataset and provide the data with the FinFET technology library.

## 6 Experiments

### 6.1 Benchmarked Models

There are many ML-based architecture-level power modeling works [20, 22, 24, 23]. We benchmark two representative existing ML-based architecture-level power models, McPAT-Calib [20] and PANDA [24], based on ArchPower. We also derive two ML-based power models, McPAT-Calib-Component and McPAT-Calib-CompGroup, based on McPAT-Calib, utilizing fine-grained component-level and power-group-level power labels. We also evaluate the classical analytical power models for comparison, including McPAT [14] and our enhanced version, McPAT-Plus.

We describe our 6 evaluated power models below. (a) McPAT [14]: A widely adopted analytical power model. Its input includes the hardware parameter and the raw event statistics. Therefore, it can also be evaluated based on our dataset. (b) McPAT-Plus: An enhanced version of McPAT. It fits a scaling factor on training data, and then scales the output of McPAT when testing. (c) McPAT-Calib [20]: It utilizes an ML model, XGBoost [11], to learn the correlation between the input feature and the final total power label. (d) McPAT-Calib-Component: An enhanced version of McPAT-Calib. It builds one ML model for each component based on per-component power labels. The per-component power predictions are summed up for total power. (e) McPAT-Calib-CompGroup: It is derived from the McPAT-Calib-Component, building one ML model for each group of each component based on per-power-group power labels. When predicting the total power, it predicts per-component power respectively and sums them up. (f) PANDA [24]: It adopts resource functions to capture the major correlation between hardware parameters and the power of each component, and multiplies it by the ML model for the final power prediction. We adopt the mean absolute percentage error (MAPE) and the correlation coefficient $R$ between label and prediction to evaluate the power modeling accuracy of the ML-based architecture-level power model.

### 6.2 Power Prediction Accuracy

#### 6.2.1 Total Power Prediction

Table 4 shows the accuracy comparison for total power prediction between our selected six models under different training scenarios. It shows that even the enhanced version of McPAT, McPAT-Plus, can not achieve a high accuracy, with MAPE over 15% and correlation coefficient $R$ lower than

| Scenario | McPAT | | | | McPAT-Plus | | | | McPAT-Calib | | | |
|---|---|---|---|---|---|---|---|---|---|---|---|---|
| | BOOM | | XiangShan | | BOOM | | XiangShan | | BOOM | | XiangShan | |
| | MAPE | R | MAPE | R | MAPE | R | MAPE | R | MAPE | R | MAPE | R |
| Balance | >100 | 0.83 | >100 | 0.85 | 18.1 | 0.83 | 29.6 | 0.85 | 8.2 | 0.98 | 33.2 | 0.73 |
| Small | >100 | 0.74 | >100 | 0.77 | 31.0 | 0.74 | 21.6 | 0.77 | 34.3 | 0.76 | 41.5 | 0.48 |
| Large | >100 | 0.83 | >100 | 0.78 | 28.2 | 0.83 | 28.3 | 0.78 | 50.6 | 0.23 | 90.0 | 0.14 |
| Average | >100 | 0.80 | >100 | 0.80 | 25.8 | 0.80 | 26.5 | 0.80 | 31.0 | 0.66 | 54.9 | 0.45 |

| Scenario | McPAT-Calib-Component | | | | McPAT-Calib-CompGroup | | | | PANDA | | | |
|---|---|---|---|---|---|---|---|---|---|---|---|---|
| | BOOM | | XiangShan | | BOOM | | XiangShan | | BOOM | | XiangShan | |
| | MAPE | R | MAPE | R | MAPE | R | MAPE | R | MAPE | R | MAPE | R |
| Balance | 6.2 | 0.98 | **14.0** | 0.97 | **6.2** | **0.98** | 15.0 | **0.97** | 6.8 | 0.97 | 19.4 | 0.9 |
| Small | 34.9 | 0.75 | 35.4 | 0.72 | 35.3 | 0.75 | 36.0 | 0.72 | **29.2** | **0.93** | **23.9** | **0.86** |
| Large | 48.9 | 0.4 | 81.4 | 0.31 | 49.2 | 0.4 | 80.5 | 0.35 | **10.4** | **0.98** | **26.3** | **0.82** |
| Average | 30.0 | 0.71 | 43.6 | 0.67 | 30.2 | 0.71 | 43.8 | 0.68 | **15.5** | **0.96** | **23.2** | **0.86** |

Table 4: Comparison between different architecture-level power models for total power prediction under different training scenarios. All MAPE values are reported as percentages. The best accuracies for each scenario are highlighted in bold.

| Component | McPAT | | | | McPAT-Plus | | | | McPAT-Calib | | | |
|---|---|---|---|---|---|---|---|---|---|---|---|---|
| | BOOM | | XiangShan | | BOOM | | XiangShan | | BOOM | | XiangShan | |
| | MAPE | R | MAPE | R | MAPE | R | MAPE | R | MAPE | R | MAPE | R |
| BP | 67.5 | 0.37 | 55.4 | 0.78 | 96.6 | 0.37 | 90.0 | 0.78 | - | - | - | - |
| ICache | 39.5 | 0.50 | 83.7 | 0.48 | 91.1 | 0.50 | 96.3 | 0.48 | - | - | - | - |
| IFU | >100 | 0.35 | >100 | 0.78 | 41.2 | 0.35 | 66.6 | 0.78 | - | - | - | - |
| RNU | >100 | 0.90 | >100 | 0.86 | >100 | **0.90** | >100 | 0.86 | - | - | - | - |
| ROB | >100 | 0.65 | >100 | 0.94 | 44.1 | **0.65** | 67.0 | **0.94** | - | - | - | - |
| ISU | >100 | 0.87 | 81.5 | 0.91 | 32.2 | 0.87 | 59.2 | 0.91 | - | - | - | - |
| Regfile | >100 | 0.73 | **49.6** | 0.81 | 43.3 | 0.73 | 70.9 | 0.81 | - | - | - | - |
| FU Pool | >100 | 0.51 | 67.9 | 0.53 | 45.0 | 0.51 | 92.8 | 0.53 | - | - | - | - |
| LSU | >100 | 0.17 | >100 | 0.84 | >100 | 0.17 | >100 | 0.84 | - | - | - | - |
| DCache | >100 | 0.71 | >100 | 0.90 | 46.8 | 0.71 | 57.1 | 0.90 | - | - | - | - |
| Other Logic | >100 | 0.70 | >100 | <0 | >100 | **0.70** | >100 | <0 | - | - | - | - |

| Component | McPAT-Calib-Component | | | | McPAT-Calib-CompGroup | | | | PANDA | | | |
|---|---|---|---|---|---|---|---|---|---|---|---|---|
| | BOOM | | XiangShan | | BOOM | | XiangShan | | BOOM | | XiangShan | |
| | MAPE | R | MAPE | R | MAPE | R | MAPE | R | MAPE | R | MAPE | R |
| BP | **1.1** | 1.00 | **12.4** | 0.90 | 1.2 | **1.00** | 12.6 | **0.90** | 1.6 | 0.99 | 24.5 | 0.78 |
| ICache | 18.7 | 0.97 | **36.2** | **0.92** | 18.2 | 0.97 | 36.3 | 0.92 | **2.2** | **1.00** | 52.8 | 0.77 |
| IFU | 14.2 | 0.42 | 16.9 | 0.86 | **13.1** | **0.48** | 16.6 | 0.87 | 36.3 | <0 | **15.5** | **0.90** |
| RNU | 45.0 | 0.68 | 15.0 | 0.92 | 56.8 | 0.64 | 15.3 | **0.94** | **43.1** | 0.79 | **14.7** | 0.86 |
| ROB | 34.3 | 0.63 | 19.4 | 0.90 | **34.1** | 0.62 | **17.6** | 0.90 | 52.0 | 0.56 | 19.7 | 0.93 |
| ISU | 24.6 | 0.91 | **38.1** | 0.81 | 24.9 | 0.91 | 51.2 | 0.78 | **21.9** | 0.84 | 43.8 | **0.93** |
| Regfile | **22.0** | 0.84 | 60.6 | 0.72 | 22.2 | **0.88** | 60.2 | 0.72 | 37.5 | 0.68 | 83.3 | **0.94** |
| FU Pool | 10.5 | 0.94 | 7.3 | 0.97 | **10.0** | **0.94** | **7.2** | **0.97** | 10.5 | 0.94 | 7.3 | 0.97 |
| LSU | >100 | <0 | 13.0 | 0.86 | >100 | <0 | 12.7 | 0.88 | **97.8** | **0.17** | **11.0** | **0.88** |
| DCache | 23.7 | 0.87 | 23.0 | 0.92 | 21.6 | 0.89 | 22.1 | 0.92 | **16.9** | **0.96** | **16.1** | **0.95** |
| Other Logic | 28.6 | 0.44 | >100 | 0.32 | **28.1** | 0.44 | **>100** | **0.87** | 44.1 | 0.34 | >100 | 0.54 |

Table 5: Comparison between different architecture-level power models for per-component power prediction under the *Balance* training scenario. All MAPE values are reported as percentages. McPAT-Calib is excluded because it does not provide per-component power information. The best accuracies for each scenario are highlighted in bold.

0.85 on all evaluations. In comparison, the ML-based architecture-level power model, McPAT-Calib, can achieve a high accuracy in the *Balance* training scenarios on BOOM. The enhanced versions of McPAT-Calib, including McPAT-Calib-Component and McPAT-Calib-CompGroup, and the advanced model PANDA can further improve the accuracy. However, in the training scenarios *Small* and *Large*, where the testing data falls out of the training data distribution, the accuracy of the existing ML-based architecture-level power model drops dramatically. It shows that our dataset can provide a comprehensive evaluation for ML-based architecture-level power models, supporting different training scenarios that can evaluate the generalization ability of models. A comprehensive evaluation demonstrates that the generalization of ML models still needs to be improved in future research.

### 6.2.2 Per-Component Power Prediction

Table 5 shows the per-component prediction accuracy of different power models. McPAT-Calib can not provide valid values because it directly predicts the final total power and does not provide

| Model | BOOM | | XiangShan | |
|---|---|---|---|---|
| | MAPE | R | MAPE | R |
| McPAT | 771 | 0.83 | 427 | 0.86 |
| McPAT-Plus | 18.3 | 0.83 | 22.5 | 0.84 |
| McPAT-Calib | **5.6** | 0.96 | **10.0** | **0.95** |
| McPAT-Calib-Comp | 6.2 | 0.95 | 11.5 | 0.94 |
| McPAT-Calib-CompGroup | 6.1 | **0.96** | 11.3 | 0.94 |
| PANDA | 7.2 | 0.95 | 11.6 | 0.92 |

Table 6: Comparison between different architecture-level power models for cross-workload prediction. All MAPE values are reported as percentages. The best accuracies are highlighted in bold.

per-component power information. It demonstrates that, besides the total power prediction, the ML-based architecture-level power model can also achieve higher accuracy compared with the traditional models for most of the components from BOOM and XiangShan.

However, it also demonstrates that ML-based architecture-level power models may also have a negative effect on some components. For example, for the RNU of BOOM CPU, the correlation coefficient $R$ of McPAT-Calib-CompGroup drops to 0.64 compared with analytical models that can achieve 0.90. It shows that our dataset can enable fine-grained evaluation for ML-based architecture-level power models and provide detailed information about the model accuracy, which shows the limitation of the existing ML-based architecture-level power models, driving potential research to improve these cases.

### 6.2.3 Cross-Workload Power Prediction

Table 6 provides our experimental results that split the training and testing scenarios based on workloads, where we adopt 8-fold cross-validation for training-testing splitting, i.e., 7 workloads for training and 1 workload for testing. The experimental results show that the ML-based power models can also demonstrate advantages over the traditional analytical model in the cross-workload scenario. It indicates that ML-based power models have great potential.

## 7   Conclusion

In this paper, we present ArchPower, the first open-source dataset for ML-based architecture-level power models. ArchPower includes 200 data samples collected from 25 CPU configurations and 8 workloads. We consider the clock-gating and integrate realistic SRAM macros for power label collection. ArchPower allows anyone to easily replicate and further improve existing architecture-level power models. We expect ArchPower to reduce the hardware barrier and enable more brilliant AI solutions in hardware design and optimizations.

## Acknowledgement

This work is supported by National Natural Science Foundation of China (NSFC) 62304192, Hong Kong Research Grants Council (RGC) YCRG Grant C6003-24Y, ECS Grant 26208723, and ACCESS – AI Chip Center for Emerging Smart Systems, supported by the InnoHK initiative of Innovation and Technology Commission of the Hong Kong Special Administrative Region Government.

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

# A   More on Dataset and Evaluation Setting

## A.1   Adopted CPU Configurations

In our dataset, we adopt 25 CPU configurations in total, including 15 configurations of BOOM CPU and 10 configurations of XiangShan CPU. Table 7 and 8 list all 25 CPU configurations adopted in our dataset.

| Hardware Parameter | B1 | B2 | B3 | B4 | B5 | B6 | B7 | B8 | B9 | B10 | B11 | B12 | B13 | B14 | B15 |
|---|---|---|---|---|---|---|---|---|---|---|---|---|---|---|---|
| FetchWidth | 4 | 4 | 4 | 4 | 4 | 8 | 8 | 8 | 8 | 8 | 8 | 8 | 8 | 8 | 8 |
| DecodeWidth | 1 | 1 | 1 | 2 | 2 | 2 | 3 | 3 | 3 | 4 | 4 | 4 | 5 | 5 | 5 |
| FetchBufferEntry | 5 | 8 | 16 | 8 | 16 | 24 | 18 | 24 | 30 | 24 | 32 | 40 | 30 | 35 | 40 |
| RobEntry | 16 | 32 | 48 | 64 | 64 | 80 | 81 | 96 | 114 | 112 | 128 | 136 | 125 | 130 | 140 |
| IntPhyRegister | 36 | 53 | 68 | 64 | 80 | 88 | 88 | 110 | 112 | 108 | 128 | 136 | 108 | 128 | 140 |
| FpPhyRegister | 36 | 48 | 56 | 56 | 64 | 72 | 88 | 96 | 112 | 108 | 128 | 136 | 108 | 128 | 140 |
| LDQ/STQEntry | 4 | 8 | 16 | 12 | 16 | 20 | 16 | 24 | 32 | 24 | 32 | 36 | 24 | 32 | 36 |
| BranchCount | 6 | 8 | 10 | 10 | 12 | 14 | 14 | 16 | 16 | 18 | 20 | 20 | 18 | 20 | 20 |
| Mem/FpIssueWidth | 1 | 1 | 1 | 1 | 1 | 1 | 1 | 1 | 2 | 1 | 2 | 2 | 2 | 2 | 2 |
| IntIssueWidth | 1 | 1 | 1 | 1 | 2 | 2 | 2 | 3 | 3 | 4 | 4 | 4 | 5 | 5 | 5 |
| DCache/ICacheWay | 2 | 4 | 8 | 4 | 4 | 8 | 8 | 8 | 8 | 8 | 8 | 8 | 8 | 8 | 8 |
| DTLBEntry | 8 | 8 | 16 | 8 | 8 | 16 | 16 | 16 | 32 | 32 | 32 | 32 | 32 | 32 | 32 |
| MSHREntry | 2 | 2 | 4 | 2 | 2 | 4 | 4 | 4 | 4 | 4 | 4 | 8 | 8 | 8 | 8 |
| ICacheFetchBytes | 2 | 2 | 2 | 2 | 2 | 4 | 4 | 4 | 4 | 4 | 4 | 4 | 4 | 4 | 4 |

Table 7: The BOOM configurations adopted in our dataset, named B1-B15. The scales of these configurations are from small to large.

| Hardware Parameter | X1 | X2 | X3 | X4 | X5 | X6 | X7 | X8 | X9 | X10 |
|---|---|---|---|---|---|---|---|---|---|---|
| FetchWidth | 4 | 4 | 4 | 4 | 4 | 8 | 8 | 8 | 8 | 8 |
| DecodeWidth | 2 | 2 | 2 | 3 | 3 | 4 | 4 | 4 | 4 | 5 |
| FetchBufferEntry | 8 | 16 | 24 | 16 | 24 | 24 | 24 | 32 | 32 | 24 |
| RobEntry | 16 | 32 | 48 | 64 | 64 | 80 | 81 | 96 | 114 | 112 |
| IntPhyRegister | 36 | 53 | 68 | 64 | 80 | 88 | 88 | 110 | 112 | 108 |
| FpPhyRegister | 36 | 53 | 68 | 64 | 80 | 88 | 88 | 110 | 112 | 108 |
| LDQ/STQEntry | 16 | 20 | 24 | 20 | 24 | 28 | 24 | 32 | 40 | 32 |
| BranchCount | 7 | 7 | 7 | 7 | 7 | 7 | 7 | 7 | 7 | 7 |
| Mem/FpIssueWidth | 2 | 2 | 2 | 2 | 2 | 2 | 2 | 2 | 2 | 2 |
| IntIssueWidth | 2 | 2 | 2 | 2 | 4 | 4 | 4 | 6 | 6 | 6 |
| DCache/ICacheWay | 4 | 4 | 8 | 4 | 4 | 8 | 8 | 8 | 8 | 8 |
| DTLBEntry | 8 | 8 | 16 | 8 | 8 | 16 | 16 | 16 | 32 | 32 |
| MSHREntry | 2 | 2 | 4 | 2 | 2 | 4 | 4 | 4 | 4 | 4 |
| ICacheFetchBytes | 2 | 2 | 2 | 2 | 2 | 2 | 2 | 2 | 2 | 2 |

Table 8: The XiangShan configurations adopted in our dataset, named X1-X10. The scales of these configurations are from small to large.

## A.2   Evaluation Metrics

We adopt the mean absolute percentage error (MAPE) and the correlation coefficient $R$, between label $Y_i$ and prediction $\hat{Y}_i$ to evaluate the power modeling accuracy of the ML-based architecture-level power model, as shown in Eq.(4).

$$MAPE = \frac{1}{n} \sum_{i=1}^{n} \left| \frac{Y_i - \hat{Y}_i}{Y_i} \right| \times 100\%, \quad R = \frac{\sum_{i=1}^{n}(\hat{Y}_i - \bar{\hat{Y}})(Y_i - \bar{Y})}{\sqrt{\sum_{i=1}^{n}(\hat{Y}_i - \bar{\hat{Y}})^2 \sum_{i=1}^{n}(Y_i - \bar{Y})^2}} \quad (4)$$

## A.3   Compute Resources

We perform our experiments on a server with Intel® Xeon® Gold 6438Y+ processor. The model evaluation is fast and efficient, taking less than one minute for each model. The memory requirement is within 10MB. Reproducing all of our results takes less than ten minutes.

### A.4 Licenses

Chipyard framework and BOOM CPU are under BSD-3-Clause. OpenXiangShan framework and XiangShan CPU are under Mulan PSL v2. Riscv-tests is under BSD-3-Clause.

## B Limitations and Future Work

While ArchPower provides the first dataset for the ML-based architecture-level power models, there are still some limitations that can be improved in future work: 1) Due to the difficulty of RTL code collection for the CPU, the diversity of architectures and the number of configurations are limited in size. Now there are only two CPU architectures with 25 configurations in our dataset. 2) The real-world single-thread workloads provided in riscv-tests are limited. Therefore, now we only include 8 workloads in our dataset for each CPU configuration.

For future work, we will have follow-up updates to ArchPower to address the two limitations above: 1) We will continue to collect new CPU architectures and provide more configurations to improve the scale of our dataset. 2) We will collect or write more real-world workloads to improve the workload diversity in our dataset.

## C Result Visualization

This section visualizes the prediction of different models on BOOM and XiangShan under different training scenarios. Each point represents a sample, and points in the same color are from the same configuration. The visualization gives a clearer comparison between different models.

Fig. 5 and 6 visualize the prediction of different models on BOOM and XiangShan under the *Balance* training scenario.

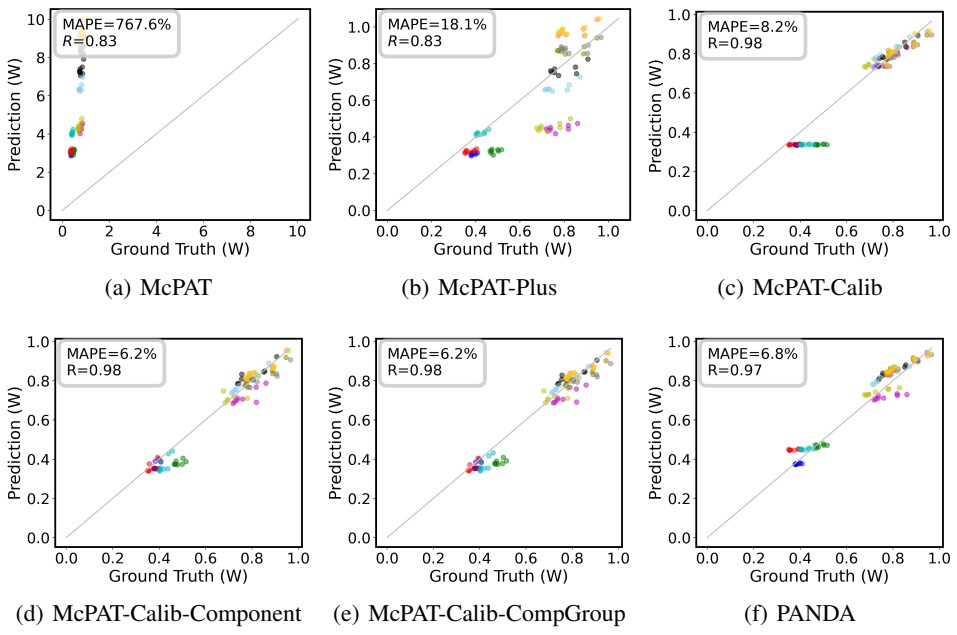

Figure 5: Predictions with different models on BOOM CPU under *Balance* training scenario.

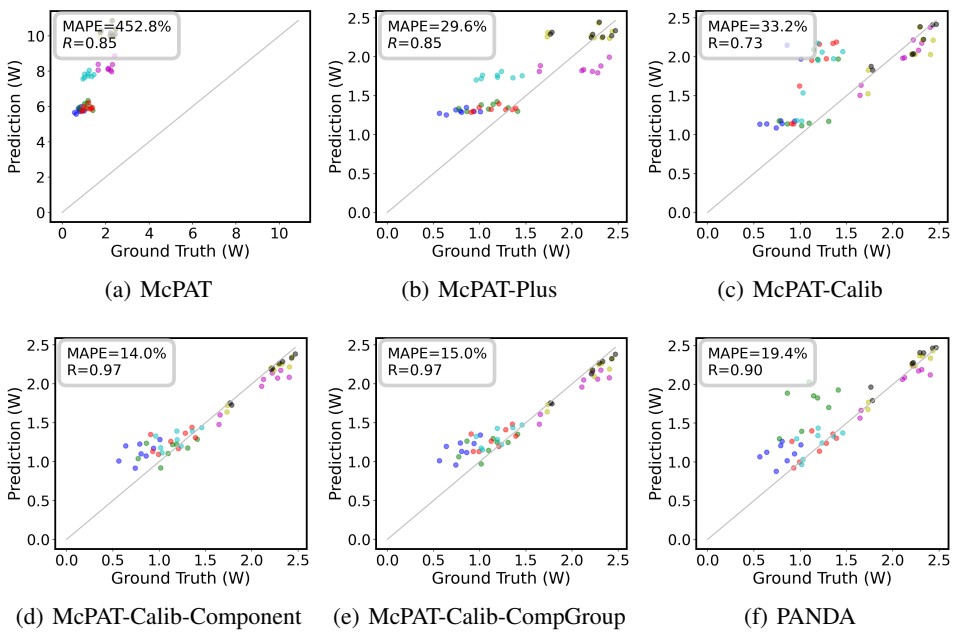

Figure 6: Predictions with different models on XiangShan CPU under *Balance* training scenario.

Fig. 7 and 8 visualize the prediction of different models on BOOM and XiangShan under the *Small* training scenario.

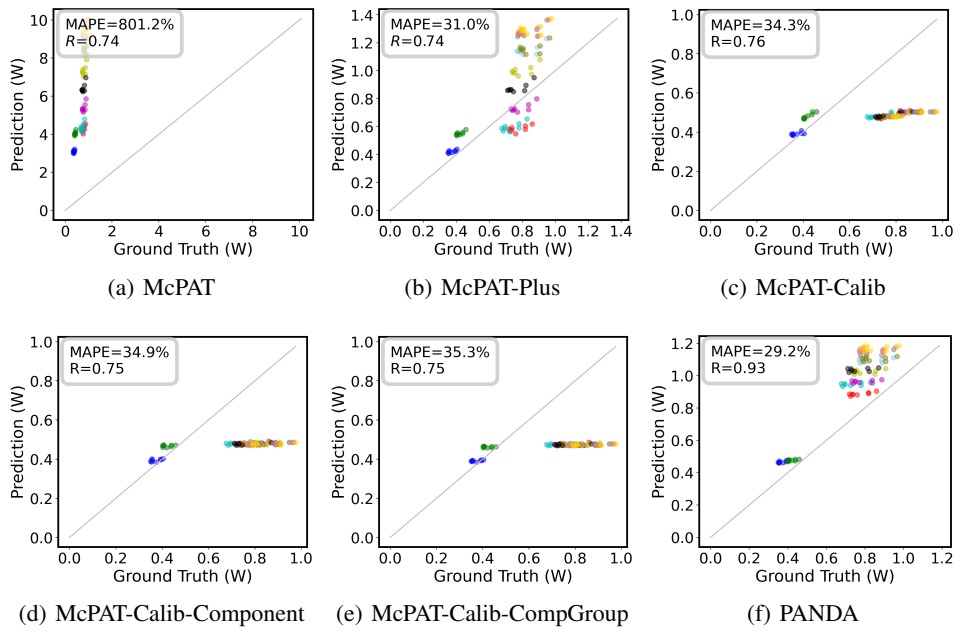

(a) McPAT      (b) McPAT-Plus      (c) McPAT-Calib

(d) McPAT-Calib-Component      (e) McPAT-Calib-CompGroup      (f) PANDA

Figure 7: Predictions with different models on BOOM CPU under *Small* training scenario.

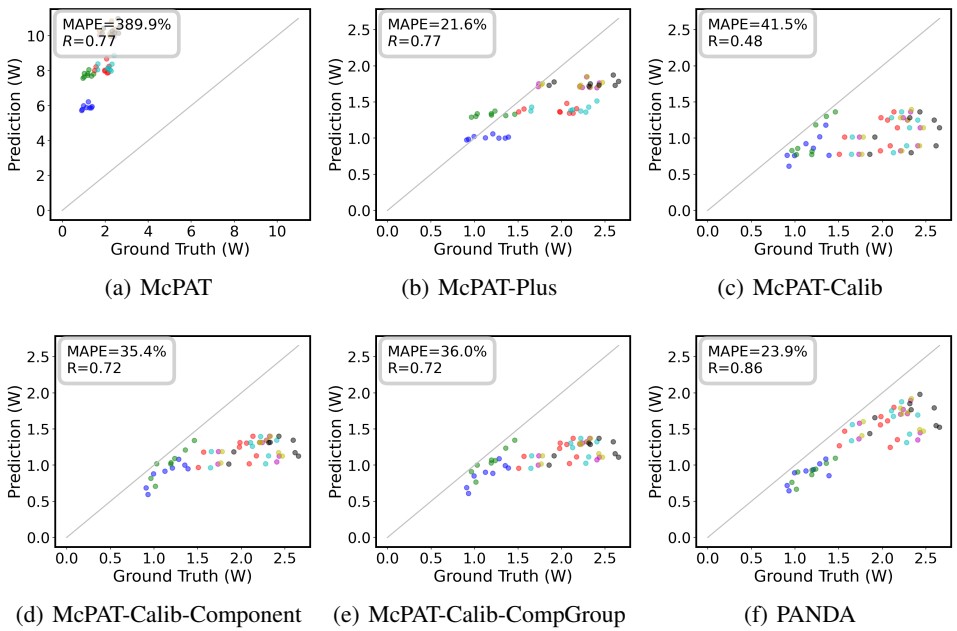

(a) McPAT      (b) McPAT-Plus      (c) McPAT-Calib

(d) McPAT-Calib-Component      (e) McPAT-Calib-CompGroup      (f) PANDA

Figure 8: Predictions with different models on XiangShan CPU under *Small* training scenario.

Fig. 9 and 10 visualize the prediction of different models on BOOM and XiangShan under the *Large* training scenario.

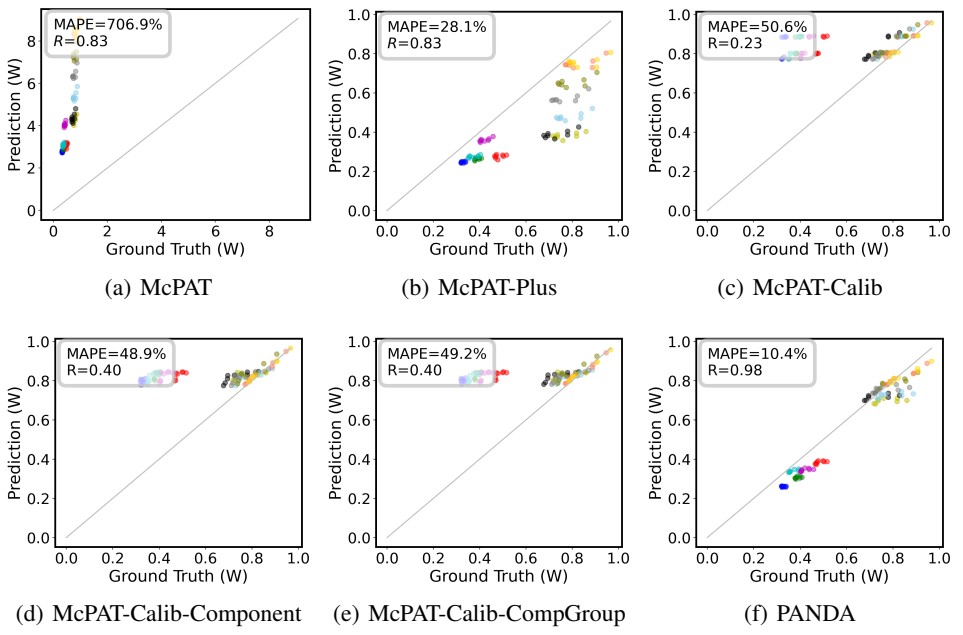

Figure 9: Predictions with different models on BOOM CPU under *Large* training scenario.

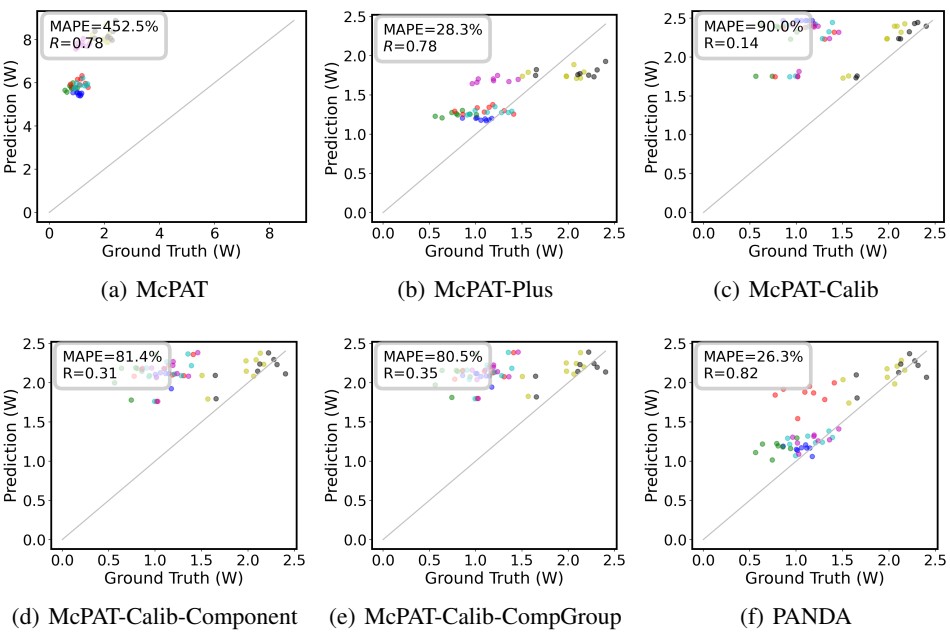

Figure 10: Predictions with different models on XiangShan CPU under *Large* training scenario.

