# OpenReview forum: "ArchPower: Dataset for Architecture-Level Power Modeling of Modern CPU Design"
_NeurIPS.cc/2025/Datasets_and_Benchmarks_Track — NeurIPS 2025 Datasets and Benchmarks Track poster_

### Official Review · Reviewer_m6wn · 2025-06-29

**Rating:** 5
**Confidence:** 3

**Summary:**

This paper proposes a CPU architecture-level power modeling dataset which includes 200 CPU data samples collected from 25 different CPU configurations and 8 different workloads. Six machine learning-based power modeling methods are also evaluated on the proposed dataset.

**Additional Feedback:**

See Limitations Weaknesses.

**Dataset Code Accessibility:**

Yes

**Dataset Code Comments:**

Authors provide the codes of the proposed benchmark and experiments in the paper on Github, the dataset is available on huggingface.

**Ethical Considerations:**

No, there are no or only very minor ethics concerns

**Final Justification:**

The responses resolve my concerns.

For the concern on the overfitting risk, the correlation among features may indeed decrease the overfitting risk. The explanation seems reasonable, CPU features could have certain patterns. Though it may imply that these features may be redundency, but it is the problem for users. Providing more features is not a disadvantage of a benchmark.

For the relationship between each CPU configuration and commercial CPUs, authors list some of the relationships and promise to improve their paper. I think it is acceptable.

For the train-test splitting, authors provide the results of train-test scenarios based on workloads, demonstrating the advantages of ML-based methods.

As a result, I update my score to 5.

**Limitations Weaknesses:**

1. Since the features have 101 dimensions and the labels have 60 dimensions, a dataset size of 200 samples seems relatively small, potentially making training challenging. However, considering the difficulty of CPU configuration design and power evaluation, this size is an acceptable starting point.

2. In Section 5.1, the authors claim that the CPU configurations are carefully selected to be similar to real-world commercial CPUs. However, I cannot find evidence supporting this. Presenting the relationship between each CPU configuration and commercial CPUs (e.g., in Tables 5 and 6) could better support the claim.

3. The authors have investigated the performance of the evaluated modelling methods under different architectures and components. Performance under different workloads is also worth exploring. Additionally, other than splitting training and testing scenarios based on the scale, investigating the split based on workloads might also make the evaluation more comprehensive.

**Strengths Contributions:**

1. This paper proposes an open-source CPU power modelling dataset, which permits reproducible research in the power modelling field.

2. The authors collect detailed features covering CPU architectures, configurations, and events. The label data is also fine-grained covering CPU components and power groups. These detailed data could benefit potential use in future researches.

3. The paper is very well written and is clearly readable.

---

> ### Author Rebuttal · Authors · 2025-07-31
>
> Thank you for your valuable feedback and insightful comments on our paper. We appreciate the time and effort you dedicated to reviewing our work. We have carefully considered all your points and provide our responses below.
>
> **Q1:**
> Since the features have 101 dimensions and the labels have 60 dimensions, a dataset size of 200 samples seems relatively small, potentially making training challenging. However, considering the difficulty of CPU configuration design and power evaluation, this size is an acceptable starting point.
>
> **A1:**
> We thank the reviewer for understanding the difficulty of collecting CPU power labels. The reviewer may have concerns about the overfitting risk because of the large feature dimensions.
> We would like to clarify the question below:
> 1. **Correlation Among Features**: Although feature has 101 dimensions, these features are information from the same CPU from different aspects. Therefore, many features are correlated with each other, not likely to incur the overfitting problem:
>    - Correlation among hardware parameters: for a large CPU, all its hardware parameters are large, while for a small CPU, all of them are relatively small.
>    - Correlation among event parameters: There are strongly correlated events among collaborating components, such as strong correlations among the number of branch predictions, the number of ICache accesses, and the number of instruction fetches.
>    - Correlation between the hardware parameter and the event parameter: for example, a large cache size will lead to a large number of cache hits and a small number of cache misses.
> 2. **Fine-Grained Information**: Although we only have 200 samples, we have 60 fine-grained labels for components and power groups for each sample. Therefore, if fine-grained modeling is carried out similarly to McPAT-Calib-CompGroup, it can be considered that our dataset has 12,000 data points, which is a very large scale. So the overfitting risk is low.
> 3. **Experimental Validation**: As our experimental data shows, even with only three configurations for training, some models can still achieve excellent results, which indicates that the data we provide is sufficient.
>
> ---
>
> **Q2:**
> In Section 5.1, the authors claim that the CPU configurations are carefully selected to be similar to real-world commercial CPUs. However, I cannot find evidence supporting this. Presenting the relationship between each CPU configuration and commercial CPUs (e.g., in Tables 5 and 6) could better support the claim.
>
> **A2:**
> We thank the reviewer for highlighting this important clarification.
> Our configurations were explicitly designed to mirror commercial CPU design philosophies, and we will enhance Tables 5-6 with direct commercial correlations in the camera-ready version. The BOOM and XiangShan configurations implement scaled versions of signature microarchitectural features from leading commercial CPUs. While not silicon replicas, they maintain functional and parametric equivalence through three key design principles.
> 1. **Front-End**: Our decode/fetch widths (4-8 instructions) directly match ARM Cortex-A78, Intel Golden Cove, and AMD Zen 4's front-end structure.
> 2. **Execution Resource Scaling**: BOOM's issue width progression (B1:1-wide → B15:5-wide) follows ARM CPU's scaling strategy. The ROB size scaling tracks commercial ROB ratios, similar to ARM Cortex-X1 and Intel Golden Cove CPU.
> 3. **Memory Hierarchy Fidelity**: LDQ/STQ entries (4-40) are scaled versions of Apple's Firestorm (32-entry) allocation policies. Cache ways (4-8) and MSHR entries (2-8) also maintain identical banking strategies to commercial designs such as ARM Cortex-A78 L1D and AMD Zen 3's L1D organization.
> 4. **Realistic Scale-Up Trend**: Our design configurations follow a realistic scale-up trend. For instance, a wider pipeline design is paired with larger caches to represent high-performance CPUs, while narrower pipelines are coupled with smaller caches to represent embedded low-power CPUs. Configurations like wide pipelines combined with small caches, which would yield impractical real-world implementations, are deliberately excluded from our design space.
> ---
>
> **Q3:**
> The authors have investigated the performance of the evaluated modelling methods under different architectures and components. Performance under different workloads is also worth exploring. Additionally, other than splitting training and testing scenarios based on the scale, investigating the split based on workloads might also make the evaluation more comprehensive.
>
> **A3:**
> We appreciate this insightful suggestion and provide new analyses. Below, we provide new experimental results that split the training and testing scenarios based on workloads, where we adopt 8-fold cross-validation for training-testing splitting.
>
> | Model                    | BOOM MAPE (%) | BOOM R | XiangShan MAPE (%)| XiangShan R |
> |---------------------|-----------|--------|---------|------|
> | McPAT              | 771       | 0.83   | 427     | 0.86 |
> | McPAT-Plus         | 18.3      | 0.83   | 22.5    | 0.84 |
> | McPAT-Calib        | 5.6       | 0.96   | 10.0    | 0.95 |
> | McPAT-Calib-Component   | 6.2       | 0.95   | 11.5    | 0.94 |
> | McPAT-Calib-CompGroup | 6.1     | 0.96   | 11.3    | 0.94 |
> | PANDA              | 7.2       | 0.95   | 11.6    | 0.92 |
>
> The experimental results show that the ML-based power models can also demonstrate advantages over the traditional analytical model in the cross-workload scenario. It indicates that ML-based power models have great potential.

---

> > ### Comment · Reviewer_m6wn · 2025-08-05
> >
> > Thanks authors response and I will update my score accordingly.

---

> > > ### Author Response · Authors · 2025-08-06
> > > **Thank you Reviewer m6wn for reviewing our responses**
> > >
> > > Thank you for reviewing our responses and updating your score. We sincerely appreciate your time, effort, and insightful feedback.

---

### Official Review · Reviewer_3C5w · 2025-07-02

**Rating:** 5
**Confidence:** 3

**Summary:**

Open dataset for architecture-level power modeling/evaluation. No such dataset exists today. Dataset has component level modeling of the CPU that is broken down into combinatorial logic, sequential logic, memory, and clock power. Power data comes from industry standard VLSI flows on 40nm. Dataset illustrates existing limitations of ML-based modeling and points to potentially valuable impact.

**Additional Feedback:**

Make the paper more readable and easier to digest.

As one example, please insert a table comparing the related works you cite, e.g., you say some datasets don’t include SRAM or clock gating. Make a table that shows that. Don’t make me do the hard work.

I’d love to see data on newer nodes, e.g., a FinFET-based flow.

I will improve my rating if I see improvements in the paper.

I changed my review to a 5.

**Dataset Code Accessibility:**

Yes

**Dataset Code Comments:**

I'm assuming yes, but didn't check (I'm not a developer, so my ability to check this is somewhat limited).

**Ethical Considerations:**

No, there are no or only very minor ethics concerns

**Final Justification:**

Good discussion with the reviewers and they are modifying their paper in accordance with suggestions to make it more obvious how they have advanced SOTA.

**Limitations Weaknesses:**

Honestly I found this paper kind of hard to follow and I have a good background in CPU architecture. It would benefit from being heavily revised in a manner that makes it easier to read.

**Strengths Contributions:**

Novel dataset. Very cool and exciting as someone who appreciates computer architecture. Huge investment to make this dataset happen, and we should greatly appreciate it extending the design space that we can explore. Evaluation also identifies the value of the dataset as it demonstrates weaknesses in existing ML-based architecture models for power estimation on OOD.

---

> ### Author Rebuttal · Authors · 2025-07-31
>
> Thank you for your valuable feedback and insightful comments on our paper. We appreciate the time and effort you dedicated to reviewing our work. We have carefully considered all your points and provide our responses below.
>
> **Q1:**
> Please insert a table comparing the related works you cite, e.g., you say some datasets don’t include SRAM or clock gating. Make a table that shows that. Don’t make me do the hard work.
>
> **A1:**
> We appreciate your suggestion of the comparison table. We provide it below:
>
> |Work    | Commercial Tech Lib | Clock Gating | SRAM Implementation | Diverse Architectures |
> |--------------------------|---------------------|--------------|---------------------|-----------------------|
> | McPAT-Calib [1]           |                     |              |                     |                       |
> | ASPDAC’23 [2]             |                     |              |                     |                       |
> | PANDA [3]                 | √                   |              | √                   |                       |
> | FirePower [4]             | √                   |              | √                   | √                     |
> | AutoPower [5]             | √                   | √            | √                   |                       |
> | ArchPower             | √                   | √            | √                   | √                     |
>
> We will add this table to the camera-ready version in the “Introduction” section.
>
>
> [1] Zhai, Jianwang, et al. "McPAT-Calib: A microarchitecture power modeling framework for modern CPUs." 2021 IEEE/ACM International Conference On Computer Aided Design (ICCAD). IEEE, 2021.
> [2] Zhai, Jianwang, Yici Cai, and Bei Yu. "Microarchitecture power modeling via artificial neural network and transfer learning." Proceedings of the 28th Asia and South Pacific Design Automation Conference. 2023.
> [3] Zhang, Qijun, et al. "PANDA: Architecture-level power evaluation by unifying analytical and machine learning solutions." 2023 IEEE/ACM International Conference on Computer Aided Design (ICCAD). IEEE, 2023.
> [4] Zhang, Qijun, et al. "Firepower: Towards a foundation with generalizable knowledge for architecture-level power modeling." Proceedings of the 30th Asia and South Pacific Design Automation Conference. 2025.
> [5] Zhang, Qijun, et al. "AutoPower: Automated Few-Shot Architecture-Level Power Modeling by Power Group Decoupling." 2025 ACM/IEEE Design Automation Conference (DAC). 2025.
>
> ---
>
> **Q2:**
> I’d love to see data on newer nodes, e.g., a FinFET-based flow.
>
> **A2:**
> We thank the reviewer for this forward-looking suggestion.
> 1. **Limited Availability of FinFET Technology**: FinFET is adopted by 14nm or newer technology nodes. However, these advanced technology libraries are not available for academia. Although there is an open-sourced ASAP 7nm library released by academia, it does not support the SRAM.  The dataset is meaningful only when adopting the high-quality commercial technology: adopting ASAP is not a good solution.
> 2. **Providing an Additional TSMC 22nm Dataset**: The most advanced technology library available to us is TSMC 22nm. We have set up the flow successfully and have been running the VLSI flow to collect the label. However, because of the time-consuming VLSI flow, we may be unable to provide the data during the rebuttal stage. We promise to provide the TSMC 22nm dataset and the evaluations in the camera-ready version, similar to what we have provided for the TSMC 40nm dataset.
> 3. **Long-Term Maintenance**: We will continuously maintain and update the ArchPower dataset. If we get access to any high-quality FinFET technology library in the future, we will update our dataset and provide the data with the FinFET technology library.
> 4. **Our Dataset as a Starting Point**: To the best of our knowledge, there is no existing open-source dataset for architecture-level power modeling. ArchPower, as the first open-source dataset in this important direction with both TSMC 40nm and 22nm, supporting 25 configurations on two architectures, is a good starting point, reducing the hardware barrier and enabling more brilliant AI solutions in hardware design and optimizations.

---

> > ### Comment · Reviewer_3C5w · 2025-08-03
> > **Thanks for the thoughtful discussion and revisions**
> >
> > Great work and really appreciate your efforts responding to my critiques and the other reviewers. I think this has improved the paper and I'll be improving my rating.

---

> > > ### Author Response · Authors · 2025-08-04
> > > **Thank you Reviewer 3C5w for thorough review of our rebuttal**
> > >
> > > Thank you for reviewing our rebuttal and increasing your rating to a 5. We sincerely appreciate your time, effort, and insightful feedback.

---

### Official Review · Reviewer_Shzo · 2025-07-03

**Rating:** 4
**Confidence:** 3

**Summary:**

This paper introduces ArchPower, the first open-source dataset specifically designed for architecture-level processor power modeling. The authors meticulously follow realistic and complex design workflows to collect CPU architectural information (features) and corresponding simulated power measurements (labels). The dataset comprises 200 CPU data samples collected from 25 distinct CPU configurations running eight different workloads. Furthermore, the authors evaluate six existing power models, including two analytical and four machine-learning (ML)-based models, using the proposed ArchPower dataset.

**Dataset Code Accessibility:**

Yes

**Ethical Considerations:**

No, there are no or only very minor ethics concerns

**Final Justification:**

I have read the response and the other reviews. I will keep my original score.

**Limitations Weaknesses:**

1.	The proposed benchmark is limited exclusively to CPU design. Extending this dataset to encompass GPU architectures and AI accelerators would significantly enhance its value, especially in an era dominated by large-scale LLMs, where the power consumption of AI accelerators critically impacts training and inference costs.
2.	Although the authors have manually gathered extensive CPU data samples and evaluated six existing power models, the novelty of both the dataset collection process and evaluation methodology remains limited. This incremental nature constrains the overall novelty and impact of the work.

**Strengths Contributions:**

1.	The proposed dataset addresses a critical need in ML-based architecture-level power prediction. Such datasets significantly benefit the research community by providing foundational resources for developing accurate and practical power models.
2.	The authors thoughtfully introduce a structured training-testing framework for ML-based architecture-level power modeling, reflecting realistic CPU design and development scenarios. This is crucial for developing robust machine-learning models that can generalize effectively to real-world applications.
3.	Extensive experiments and evaluations conducted by the authors effectively demonstrate the utility and applicability of the ArchPower benchmark, providing valuable insights into the performance of various power modeling techniques.

---

> ### Author Rebuttal · Authors · 2025-07-31
>
> Thank you for your valuable feedback and insightful comments on our paper. We appreciate the time and effort you dedicated to reviewing our work. We have carefully considered all your points and provide our responses below.
>
> **Q1:**
> The proposed benchmark is limited exclusively to CPU design. Extending this dataset to encompass GPU architectures and AI accelerators would significantly enhance its value, especially in an era dominated by large-scale LLMs, where the power consumption of AI accelerators critically impacts training and inference costs.
>
> **A1:**
> We agree that GPUs and AI accelerators are important. However, we do not include them currently because of these reasons:
> 1. **Absence of High-Quality Representative Open-Source Implementation**: There are no open-source GPU or AI accelerator implementations that approach the level of industrial-level products. Released materials typically include only high-level ISA specifications without microarchitectural details, or partial core designs lacking full memory hierarchies or power management infrastructures. This prevents acquiring cycle-accurate event traces and high-quality power labels essential for credible ML research.
> 2. **Architectural Divergence from CPU Research**:
> Unlike CPUs, which employ centralized out-of-order execution, unified cache hierarchies, and scalar execution units, GPUs and accelerators operate through massively parallel SIMT architectures where warp scheduling replaces conventional instruction-level parallelism. For CPU, the power modeling focuses on caching behaviors and complex control-flow logic. GPU and accelerator introduce unique power characteristics: tensor cores and MAC arrays exhibit non-linear energy scaling under sparse computations, partitioned HBM memories demonstrate access patterns decoupled from traditional caching behaviors, requiring fundamentally new methodologies beyond our CPU-focused approach.
> 3. **CPU Research Criticality**: General-purpose CPUs still remain ubiquitous across devices. CPUs still account for approximately half of the market share, and their power consumption remains a critical issue. For instance, in some data centers, CPU power consumption can constitute over 40% of total energy usage. Our dataset tackles the long-standing absence of datasets for this domain.
>
> ---
>
> **Q2:**
> Although the authors have manually gathered extensive CPU data samples and evaluated six existing power models, the novelty of both the dataset collection process and the evaluation methodology remains limited. This incremental nature constrains the overall novelty and impact of the work.
>
> **A2:**
> We understand your concern about the novelty, but we think the data collection flow should be rigorous. We adopt a standard VLSI design flow used in industry, rigorously adhering to industrial practices to ensure data quality.
>
> Our novelty lies in other perspectives.
> 1. **Unprecedented Dataset Scope**: We provide the first public dataset featuring in such an important domain.
> 2. **Implementation Advancement**: Our work introduces innovation compared to prior efforts, as we incorporate synthesized CPU configurations and include implementations of SRAMs and clock gating.

---

> > ### Comment · Reviewer_Shzo · 2025-08-09
> >
> > Thanks for the detailed response. I will maintain my original score.

---

### Official Review · Reviewer_TiML · 2025-07-07

**Rating:** 4
**Confidence:** 4

**Summary:**

Power evaluation is a critical but time-consuming task in large-scale IC design, particularly for modern CPUs. While ML-based models offer improvements, the lack of available datasets is a major challenge. This work introduces ArchPower, the first open-source dataset for architecture-level CPU power modeling. It includes 200 data samples from 25 CPU configurations running 8 workloads, with over 100 architectural features and fine-grained power information for each sample.

**Dataset Code Accessibility:**

Yes

**Ethical Considerations:**

No, there are no or only very minor ethics concerns

**Final Justification:**

After reviewing the rebuttal, I acknowledge the authors’ clarification of the simulation–product gap common in modern CPU design. Nevertheless, the dataset size still feels insufficient for a NeurIPS contribution, so I maintain my original rating.

**Limitations Weaknesses:**

- The datasets are derived from two open-source CPU architectures, BOOM and XiangShan. It is unclear how these datasets benefit modeling and prediction for commercial CPUs like Intel x86 and ARM. If they do not, what are the gaps and challenges in applying the benchmark to commercial CPUs?
- Some Mean Absolute Percentage Errors (MAPEs) are still high, exceeding 20% for certain sub-components. It is suggested to identify and explain the reasons for these high errors.
- Both the event statistics and ground-truth power data are generated by simulators, but the paper does not clarify the differences between simulated data and actual physical device data.

**Strengths Contributions:**

+ The proposed power modeling framework is comprehensive and outperforms existing methods.
+ The power data includes both total circuit power and multiple domain values, enabling fine-grained on-chip power estimation.
+ The workflow for data collection, model training, and evaluation is clear, reasonable, and complete.

---

> ### Author Rebuttal · Authors · 2025-07-31
>
> Thank you for your valuable feedback and insightful comments on our paper. We appreciate the time and effort you dedicated to reviewing our work. We have carefully considered all your points and provide our responses below.
>
> **Q1:**
> The datasets are derived from two open-source CPU architectures, BOOM and XiangShan. It is unclear how these datasets benefit modeling and prediction for commercial CPUs like Intel x86 and ARM. If they do not, what are the gaps and challenges in applying the benchmark to commercial CPUs?
>
> **A1:**
> We appreciate the concern about generalizability to commercial CPUs. We think our dataset is a good starting point.
>
> 1. **Access Barrier to Commercial CPU**: Commercial CPUs, such as Intel and ARM CPUs, represent critical intellectual property for these companies. No entity, including the companies themselves, would disclose these CPU designs. Therefore, we consider using BOOM and XiangShan for collecting datasets to be a good starting point.
> 2. **Microarchitectural Similarity**: Open-source designs like BOOM and XiangShan adopt out-of-order execution architectures consistent with modern high-performance commercial CPUs such as Intel and ARM. This ensures a high degree of similarity at the microarchitectural level with similar component build-up, including Instruction Fetch Unit, ICache, DCache, Branch Predictor, etc. Therefore, for academic research, BOOM and XiangShan serve as excellent substitutes for commercial CPUs.
> 3. **RISC-V Being Increasingly Important**: These CPUs use RISC-V – an ISA gaining rapid commercial adoption (e.g., SiFive, Alibaba T-Head). RISC-V CPUs hold significant potential today, have experienced rapid development in recent years, and are poised to play an increasingly important role in future markets. Consequently, our choice of these two CPUs will be highly representative moving forward.
> 4. **Pioneering Contribution in Open Data**: We establish the first open-source benchmark enabling ML research for architectural power prediction, bridging a critical gap where no prior public datasets exist.
> 5. **Similar Modeling Challenges**: Commercial CPUs also rely on architectural-level performance simulators for early-stage exploration and development. Therefore, from a design methodology perspective, the challenges in power modeling for commercial CPUs—and the need to bridge the gap between simulation and actual design—are similar to those faced with open-source CPUs. For example, [1] analyzed power modeling for IBM CPUs and identified that the gaps between power modeling and real design primarily include high-level logic abstraction and configuration inconsistencies. These findings align with the issues we observed in power modeling for open-source CPUs.
>
> [1] Xi, Sam Likun, et al. "Quantifying sources of error in McPAT and potential impacts on architectural studies." 2015 IEEE 21st International symposium on high performance computer architecture (HPCA). IEEE, 2015.
>
> ---
>
> **Q2:**
> Some Mean Absolute Percentage Errors (MAPEs) are still high, exceeding 20% for certain sub-components. It is suggested to identify and explain the reasons for these high errors.
>
> **A2:**
> We thank the reviewer for highlighting this point. The reasons for these high errors are summarized below:
> 1. **Few-Shot Learning Challenge**: Due to the extremely high cost associated with acquiring CPU power data, architectural-level power modeling necessitates few-shot learning. For some more complex components, existing methodologies fail to learn the correlation between features and labels effectively with such limited samples, resulting in higher error rates, as discussed in PANDA [1].
> 2. **Simulator-RTL Discrepancy**: Architectural simulators abstract low-level circuit behaviors, creating inherent gaps versus RTL. There exists an inevitable discrepancy between the events captured by architectural-level simulators and the actual RTL implementations. Current approaches ignore this discrepancy, leading to modeling inaccuracies for specific components.
>
> But we do not think the high error rate is the problem with this work. We’ve rigorously replicated 6 state-of-the-art methods, creating the only benchmark for fair comparison. Our high MAPE components reveal open research challenges.
>
> [1] Zhang, Qijun, et al. "PANDA: Architecture-level power evaluation by unifying analytical and machine learning solutions." 2023 IEEE/ACM International Conference on Computer Aided Design (ICCAD). IEEE, 2023.
>
> ---
>
> **Q3:**
> Both the event statistics and ground-truth power data are generated by simulators, but the paper does not clarify the differences between simulated data and actual physical device data.
>
> **A3:**
> We understand the concern about simulation validity. However, we think simulation is the best way for data collection.
> 1. **Cost-Prohibitive Post-Silicon Data Collection**: Taping out for every configuration of each CPU is impractical, incurring prohibitive costs. Even in industry, it is infeasible to tape out every one of such diverse configurations.
> 2. **Granular Measurement Impossibility**: Measuring fine-grained power on physical devices is difficult; they cannot provide fine-grained measurements of power for individual components and power groups, which fails to meet our requirements.
> 3. **Industry-Level Simulation Flow**: We also emphasize that because we have access to the actual netlist implementation, adopt industry-standard commercial EDA tools and technology libraries, and have implemented actual SRAMs and clock gating, the gap between our power simulation results and the final post-tape-out power consumption is minimal.

---

> > ### Comment · Reviewer_3C5w · 2025-08-03
> > **Agree with authors A3**
> >
> > I would echo everything authors say in A3. Actual CPU tape-outs are incredibly expensive and that's why the data is difficult to obtain.

---

### Note · Authors · 2025-08-14

We sincerely thank all the reviewers and the AC for the time and effort dedicated during the rebuttal period. We are delighted to see that several reviewers increased their scores during the rebuttal. Now all reviewers have assigned a positive rating to our paper, indicating a significant recognition of our open dataset on CPU power modeling. Reviewers particularly highlight multiple aspects of our dataset:

1. The significance of our open dataset for CPU power modeling is recognized by **Reviewer Shzo** (*addresses a critical need*), **Reviewer 3C5w** (*very cool and exciting*), and **Reviewer m6wn** (*permits reproducible research*).
2. The comprehensiveness of our dataset has been acknowledged by **Reviewer TiML** (*enabling fine-grained power estimation*), **Reviewer 3C5w** (*extending the design space*), and **Reviewer m6wn** (*collect detailed features, label data is also fine-grained*).
3. The thoroughness of our evaluation is also recognized by **Reviewer TiML** (*evaluation is complete*), **Reviewer Shzo** (*reflecting realistic scenarios, extensive experiments and evaluations*), **Reviewer 3C5w** (*identifies the value of the dataset*), and **Reviewer m6wn** (*I will update my score [after we provided cross-workload evaluation in rebuttal]*).
4. The clarity of our writing has also been acknowledged by **Reviewer TiML** (*workflow is clear*), **Reviewer 3C5w** (*has improved the paper [after we further improved the writing in rebuttal]*), and **Reviewer m6wn** (*very well written*).

We expect our paper to support more researchers to work on such an important area and facilitate the design of cutting-edge CPUs.

---

### Decision · Program_Chairs · 2025-09-18

**Decision:**

Accept (poster)

**Comment:**

## Summary

This work proposes a dataset to estimate architecture-level power consumption via ML models.
It provides 200 realistic CPU data samples based on 25 CPU configurations and 8 workloads.

## Strengths

+ The authors collect detailed features covering CPU architectures, configurations, and events. The label data is also fine-grained covering CPU components and power groups. These detailed data could benefit potential use in future researches.
+ The proposed dataset addresses a critical need in ML-based architecture-level power prediction. Such datasets significantly benefit the research community by providing foundational resources for developing accurate and practical power models.


## Weaknesses

- Limited dataset size and generalizability. It is not clear how the study can be extended to commercial CPUs such as Intel x86 and ARM (TiML), and GPU architectures (Shzo).
- [Addressed] The number of samples (200) is also limited and very sparse as it contains 101 features and 60 possible labels (m6wn). However, this last one has been mitigated by the response from the authors, clarifying that the features are possibly correlated and redundant to capture different aspects of the same CPU and the labels are fine-grained information.
- [Addressed] Although the authors have manually gathered extensive CPU data samples and evaluated six existing power models, the novelty of both the dataset collection process and evaluation methodology remains limited. This incremental nature constrains the overall novelty and impact of the work. (Shzo)
- [Addressed] Event statistics and ground-truth power data are generated by simulators, but it's not clear how they would become on real physical devices (however, this is due to alternatives being prohibitively expensive)


## Summary of the Discussion and Final Decision

The reviewers agree that this work is valuable and provides a good contribution to the track. The authors addressed most of the reviewers' concerns during the rebuttal, leaving partially unaddressed the first one pointed out due to access barriers of commercial CPUs (especially related to IP) and the use of simulators (due to the high cost of the alternative, as also backed up by reviewer 3C5w).